# Diurnal pattern of respiration in corals and algae and its implications for gross primary production quantification

Yvonne Sawall[1,2]*, Roderick Bakker[1], Natalia E. Padillo-Anthemides[3], Nicole Adamson[4]

**1** Bermuda Institute of Ocean Sciences (BIOS), 17 Biological Station, St. George's, Bermuda, **2** School of Ocean Futures (SOF), Arizona State University (ASU), 777 E University Drive, Tempe, Arizona, United States of America, **3** University of Florida, Whitney Laboratory for Marine Bioscience, St. Augustine, Florida, United States of America, **4** University of California San Diego, California, United States of America

* yvonne.sawall@bios.asu.edu

## Abstract

Mitochondrial respiration ($R$) and gross photosynthesis ($GP$) are crucial components of the energy and carbon budgets of photosynthesizing organisms in coral reefs. This study investigates the diurnal and seasonal patterns of $R$ in common reef algae and corals, examining the relationship between $R$ and photosynthesis. Additionally, it evaluates discrepancies between daily $R$ and $GP$ calculations based on diurnal variations versus constant nighttime $R$, latter being the more traditional approach. We collected three coral species (*Montastrea cavernosa*, *Porites astreoides*, and *Diploria strigosa*) and three algal species (*Caulerpa verticillata*, *Ceramium nitens*, and *Laurencia obtusa*) from a Bermuda reef in fall (October) and measured their metabolic rates in a controlled outdoor mesocosm environment. Diurnal patterns of photosynthesis were measured under natural sunlight, and respiration was measured at different times by covering the incubations with a black sheet. Measurements were repeated with re-collected corals in spring (April) and summer (July). Our findings reveal pronounced diurnal patterns in $R$ for both corals and algae, with peak $R$ in the afternoon, lagging behind peak $GP$ by 1–3 hours. Seasonal analysis showed the highest $R$ in summer and the lowest in fall, correlating with temperature and light intensity variations. The study indicates that traditional models, assuming constant nighttime $R$ throughout the day, underestimate daily $R$ and $GP$ rates by an average of 14% and 13%, respectively, and by 23% and 18% at a maximum. These results highlight the need to incorporate the dynamic nature of respiration into our understanding of energy and carbon fluxes in reef organisms. As metabolic energy availability is crucial for organism resilience, improved estimates of $R$ and $GP$ are essential for predicting organism survival in a changing environment.

**Data availability statement:** All relevant data are within the paper and its Supporting information files.

**Funding:** YS supported by BIOS Cawthorn Innovation Fund project (2019-2021). RB supported by the Erasmus program for student exchange. NPA and NA supported by the National Science Foundation REU Program at BIOS, Award# OCE-1757475. The funders had no role in study design, data collection and analysis, decision to publish, or preparation of the manuscript.

**Competing interests:** The authors have declared that no competing interests exist.

## 1. Introduction

Hermatypic corals and algae are the predominant benthic organisms in coral reefs. Through respiration and photosynthesis, they play a significant role in carbon and energy flow within coral reef ecosystems [1]. Like in most organisms, mitochondrial respiration of corals and algae is the metabolic process that transforms stored energy from organic carbon sources like carbohydrates, lipids, and proteins into adenosine triphosphate (ATP), the primary energy carrier within cells. ATP fuels essential cellular activities, including anabolic processes (macromolecule synthesis), gene regulation, transmembrane transport, and cell signaling.

Key energy expenditures for corals and algae include cell maintenance, defense mechanisms, biomass growth, calcification (in corals and some algal species), mucus production, and reproduction [2–4]. These energy-demanding processes compete with each other, especially during energy scarcity or when environmental stressors increase the demand for mitigation or repair mechanisms [5,6]. The main source of energy in many coral species [2,7] and the sole source of energy in algae is photosynthesis. As coral reefs face growing exposure to environmental stressors such as pollution, heat waves, and ocean acidification, it becomes crucial to accurately measure energy acquisition and utilization [8]. This will improve understanding and predictions of coral resilience as well as potential shifts in the flow of energy and carbon through reef ecosystems.

The amount of energy gained through photosynthesis and the total amount of energy utilized (i.e., respiration rate) is commonly measured via respirometry. Here, an organism is placed inside an air-tight incubation chamber, and the change in $O_2$ or $CO_2$ over time (i.e., over 30 minutes) is recorded (e.g., [9–11]). An increase of $O_2$ (or a decrease of $CO_2$) during light conditions represents the net photosynthesis rate (NP), and a decrease of $O_2$ (or an increase of $CO_2$) during dark (or nighttime) conditions represents the respiration rate (R). Gross photosynthesis (GP), representing total light-dependent energy acquisition, is calculated by summing NP and R. However, there can be considerable uncertainty in estimating GP, depending on the conditions under which R is measured. Daytime respiration is light-dependent, meaning longer durations of illumination and higher light intensities can increase R, as observed in corals [12–15] and algae [16,17]. For instance, Edmunds and Davies [12] found coral respiration rates to increase by up to 58% following an 80-minute illumination period compared to pre-illumination rates. This effect can extend over longer periods, as shown by Falter et al. [18] in a Hawaiian field experiment, where high daytime photosynthesis rates on sunny days were followed by elevated nighttime respiration rates, while low photosynthesis rates on overcast days corresponded with reduced nighttime respiration rates. This phenomenon likely occurs because oxygen, carbohydrates, and lipids produced during photosynthesis directly fuel respiration.

Although it is well recognized that R is light-dependent, studies that calculate GP do not typically consider the dynamics of R. These studies can roughly be categorized into 4 main approaches:

*(i) NP* is measured under natural light conditions throughout the day, with R measured at night. Nighttime R is then added to NP rates to calculate hourly, maximum,

or daily integrated *GP*. (e.g., [19–23]). This approach may underestimate *GP* by incorrectly assuming that daytime *R* equals nighttime *R*. *(ii)* *NP* and *R* are measured in the laboratory to construct photosynthesis-irradiance (P-I) curves, starting with no light and incrementally increasing light levels until *NP* is light-saturated. *R*, measured as the first rate of the P-I curve, is then added to the *NP* rates at each light level to calculate the respective *GP* rates (e.g., [24–26]. Here, again, *GP* may be significantly underestimated, because dark-acclimated *R* may be lower than expected *R* at increasing light levels. *(iii)* *NP* and *R* are measured under specific light conditions (constant light in the lab or natural light at a specific time of day), where *R* measurement immediately follows *NP* measurement. *GP* is then calculated from these rates, assuming *R* continues at a similar rate as during the prior light period (e.g., [11,23,27–30]). This approach provides *GP* rates closest to the true *GP* at the given light intensity, but does not allow to accurately extrapolate to daily rates. *(vi)* In a fourth approach, *R* is derived from $O_2$ consumption immediately after turning off the light using fast-responding $O_2$ microsensors at the surface of the organism. Within the first few seconds, $O_2$ consumption is particularly high, resulting in 6- to 25-fold higher $O_2$ consumption rates than using traditional respirometry approaches described under approaches *i* to *iii* [13,31,32]. This is because these rates include not just mitochondrial respiration, but also the various other $O_2$-consuming processes that occur during light conditions, such as photo-respiration and photo-protective processes, that cease rapidly after darkening [16]. The Mehler reaction has been identified as the main $O_2$-consuming photo-protective mechanism in corals, which consumes ATP instead of producing it [33]. Therefore, this approach can lead to an overestimation of the total energy gain (*GP*) and energy conversion into ATP (mitochondrial *R*).

To calculate more accurately in-situ energy and carbon budgets for corals and algae and ultimately for reef communities, we need to determine daytime *R* under natural light conditions. This data is not yet available, which also leads to a lack of understanding of how well commonly reported *GP* rates represent "true" rates of energy gain through photosynthesis. Acquiring more accurate knowledge of energy gain (*GP*) and conversion (*R*) will improve our understanding of the capacity of reef photosynthesizers to cope with environmental stress that typically comes with an energy trade-off. In this study, we addressed the following objectives:

I.  Investigate diurnal patterns of respiration in common reef primary producers and their relationship to light intensity/photosynthesis.

II.  Investigate seasonal variability of respiration and potential drivers.

III.  Evaluate the potential difference between daily *R* and daily *GP* based on the diurnal variation in *R* versus a scenario with constant nighttime *R*.

## 2. Material and methods

### 2.1 Study species

Experimental corals and algae were collected from a rim reef (Hog Reef) in Bermuda at 4-5m depth in September 2020 (Collection permit provided by the Department of Environment and Natural Resources, Bermuda Government, #2020092209). Nine coral colonies (~∅15cm) of each of the following 3 species were removed from the reef with a hammer and chisel: *Montastrea cavernosa*, *Porites astreoides,* and *Diploria strigosa.* All species are dominant reef builders in Bermuda and are highly abundant throughout the Western Atlantic. *M. cavernosa* usually harbors a diverse zooxanthellae (*Symbiodiniaceae*) community of clade A, C, and D, with clade C being most dominant. *P. astreoides* harbors a consortium of clade A types [34], and *D. strigosa* harbors clade B [35]. Algae were collected by removing a total of 27 small "rocks" (calcium carbonate structures) covered by algae, each rock being dominated by one of the three common algal species: *Caulerpa verticillata*, *Ceramium nitens*, and *Laurencia obtusa* (9 rocks per species; Fig. S1 in S4 Data). Other species on the rocks contributed less than 5%, including some fleshy algae species, turf algae, crustose coralline red algae, small ascidians, and sponges. These organisms were not removed, as their impact was relatively minor, and attempting to do so would have risked accidentally dislodging the algae of interest.

## 2.2 Coral and algae maintenance

Corals and algae were left in a large outdoor water basin (holding tank) of the mesocosm facility at the Bermuda Institute of Ocean Sciences (BIOS), equipped with a pond pump for water mixing and supplied with flow-through seawater from the adjacent Reach. Light intensity was adjusted to light levels at the site/ depth of origin (~60% of surface irradiance) by covering the basins with a fly screen. Measurements of metabolic rates (*fall* measurements) started 5 days after collection, first with algae (1st to 15th of October 2020) and then with corals (18th to 30th of October 2020). After completion of measurements, each coral was glued (underwater epoxy) to a plexiglass tile and returned to its site of origin by attaching the tile to a cinderblock cemented to the reef. Corals were recollected in March 2021 for the *spring* measurements (7th to 29th April 2021) and again in July 2021 for the *summer* measurements (12th to 27th of July 2021; Table 1). Measurements started within 1 week after collection. No algae were used for the *spring* and *summer* measurements.

## 2.3 Experimental setup of incubations

In addition to the coral holding tank in the BIOS mesocosm facility, three 500-L basins were prepared to be used as water baths, where specimens were incubated in a total of 9 incubation chambers (3 chambers/ basin). Each water bath also acted as a reservoir for fresh seawater during flushing of the incubation chambers in between incubation periods (see below). Like the holding tank, each water bath was supplied with flow-through seawater and hosted a large pond pump to facilitate water mixing. Light levels were adjusted by covering them with a fly screen, and the temperature was kept constant (±1°C) using aquarium heaters and chillers (TK-6000, TECO US). Temperature was monitored by using a hand-held thermometer and by deploying temperature loggers (Pendant HOBO loggers) in each basin.

The incubation chambers consisted of 19.5-L aquaria with Plexiglas lids for sealing. Each chamber was equipped with a HOBO logger for temperature and light measurements every 10 seconds. Only 7 HOBO loggers were available, which rotated between the 9 incubation chambers for spot-checking that the temperature remained constant during incubations and to measure light intensity throughout the day. Light was recorded in lux and later converted to PAR [$\mu$mol photons $m^{-2}$ $s^{-1}$] using a logger-specific conversion derived from an intercalibration of the HOBO loggers with an underwater PAR sensor (LI-192, LiCor Biosciences) at experimental conditions. Intercalibrations followed an exponential decay relationship [36] and were repeated in each season.

For oxygen measurements, optical oxygen sensor spots (OXSP5, Pyro-Science GmbH, Germany) were glued into each incubation chamber, and oxygen was measured with an oxygen meter through the chamber glass (FireSting, Pyro-Science GmbH) either continuously (*fall*) or in the beginning and at the end of an incubation period for a duration of 30 seconds per chamber (*summer*). Discontinued measurement in *summer* was due to an insufficient amount of oxygen meters available, resulting in having to measure oxygen concentration in one chamber after the other. In *spring*, battery-run optical oxygen sensors (MiniDOT, PME, USA), fully submerged in the chambers, were used to record oxygen concentration once every minute. The comparability of the 2 types of oxygen sensors (FireSting vs MiniDOT) was tested by determining the change of oxygen over time inside two incubation chambers with both sensors simultaneously

**Table 1. Environmental conditions during incubations.** Temperature is provided as minimum (Min), maximum (Max), and mean. Photosynthetic active radiation (PAR) is the mean (SE), and solar radiation is the mean (SD).

| Taxa | Season | Date of incubations | Temperature [°C] | | | PAR (aquaria) | Solar radiation (weather station) |
|------|--------|---------------------|------|------|------|----------------|-----------------------------------|
| | | | Min | Max | Mean | [mol photons $m^{-2}$ $d^{-1}$] | [Watt $m^{-2}$ $d^{-1}$] |
| Algae | Fall | 1-15 Oct. 2020 | 25.0 | 27.5 | 26.3 | 7.25 (1.67) | 16.89 (4.33) |
| Coral | Fall | 17-29 Oct. 2020 | 24.0 | 27.0 | 25.5 | 7.02 (1.00) | 18.05 (5.41) |
| Coral | Spring | 7-29 Apr. 2021 | 20.0 | 22.5 | 21.3 | 12.55 (0.75) | 25.09 (6.34) |
| Coral | Summer | 12-27 Jul. 2021 | 27.5 | 30.5 | 29.0 | 10.09 (0.95) | 24.24 (8.27) |

(Fig S2 and S3 in S4 Data). No significant difference was found between the 2 sensors (Fig S3 in S4 Data). To facilitate water movement and ensure a homogenous distribution of oxygen in the chambers, small submersible aquarium pumps were used (DOMICA Mini submersible water pump).

## 2.4 Incubation schedule for net photosynthesis (*NP*) and respiration (*R*) measurements

*NP* and *R* rates were measured of individual corals or algae "rocks" placed inside incubation chambers based on changes in dissolved oxygen concentrations. Incubations started ~30 minutes before sunrise and ended ~1 hour after sunset, with the exception of some days when incubations were carried out for 24 hours to measure night-time respiration, as well. Incubations lasted for 30 minutes. Water was exchanged every 1–4 hours: every 2 hours in the morning and late afternoon, and every hour between 10:00 and 16:00 hours to avoid extreme oxygen oversaturation. In the night, flushing was conducted every 3–4 hours. Day-time respiration was measured in between light incubations by covering the basins with a thick black tarp, allowing no light to reach the incubation chambers for a duration of one incubation period (no light was confirmed by light measurements). This was done with one of three basins at a time, alternating between basins every hour (e.g., basin #1: 8:00–8:30, basin #2: 9:00–9:30, basin #3: 10:00–10:30, basin #1: 11:00–11:30 etc.), keeping the total amount of dark periods per basins and hence per incubated individuum at a minimum (3–4 per day). This is important considering that photosynthesis rates affect respiration rates. Eight to nine specimens of corals or algae were measured per day (2–3 specimens per species), while distributing them randomly between basins and incubation chambers. Each specimen was measured 3 times per season. To account for $O_2$-consuming or producing processes in the water column, 4–8 day-long coral-/ algal-free control incubations were run per season, confirming negligible rates of water column processes.

## 2.5 Metabolic rate standardization and calculation

Metabolic rates of corals were standardized to the 2-D surface area of coral colonies and to the 2-D surface area of algae assemblages, which were determined by taking pictures from above including a ruler for reference. The pictures were imported into the software ImageJ to calculate the surface area (Fig S1 in S4 Data). Although metabolic rates of algae are more commonly standardized to biomass, we decided to use algal 2D surface area to (*i*) make it more comparable to the coral metabolism rates and, (*ii*) because community/ reef-scale metabolic rate measurements typically rely on measurements standardized to surface area.

Hourly rates of *NP* or *R* were calculated for each incubation period, based on the change of oxygen concentration ($\Delta O_2$ in µmol $L^{-1}$) over time, neglecting data points during flushing. Changes of oxygen concentration (slope) were corrected for water volume in the incubating chambers and standardized to duration of incubation period and surface area of the organism (e.g., [10]). Hourly *GP* was calculated by summing up hourly *NP* and *R* at any given time of the day from the fitted curves that describe the diurnal pattern (see curve fitting below; Fig 1). Daily rates were then calculated by integrating the hourly rates over the respective time periods: over 24 hours for respiration and from sunrise to sunset for gross photosynthesis (Fig 1, filled areas). These rates were annotated *R*new and *GP*new, respectively, with "new" indicating that the rates incorporate measured daytime *R*. To compare *R*new and *GP*new with rates that are calculated in the more traditional way of using nighttime *R* (approach 1 in the introduction), (*i*) daily *R*trad (trad = traditional) was calculated by multiplying average hourly nighttime *R* by 24 hours and (*ii*) daily *GP*trad was calculated by integrating the hourly *GP* rates that were calculated from hourly *NP* and averaged hourly nighttime *R* between sunrise and sunset (Fig 1, red dashed lines/ areas). Ratios of daily gross photosynthesis to respiration were calculated by dividing *GP*new with *R*new and *GP*trad with *R*trad.

## 2.6 Curve fitting and error propagation

A generalized additive model (GAM) was applied to hourly rates of *R* and *NP* versus time to describe their diurnal patterns, using the software "R" (package: mgcv 1.8; [37]) and the method "REML". The GAM was done with cumulative data

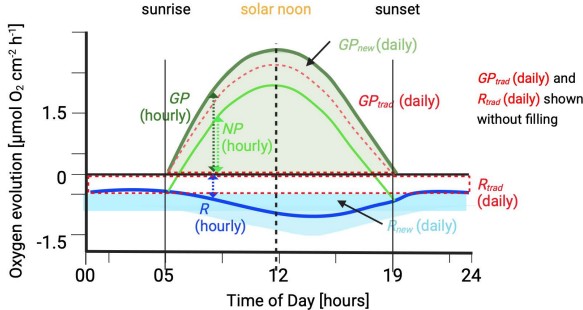

**Fig 1. Schematic representation of measured and derived metabolic rates.** All daily rates have the annotation "new" (= approach incorporation hourly $R$ measured in the daytime) or "trad" (= using the more traditional approach based on hourly nighttime $R$). Dashed arrows = hourly rates of $GP$ (dark green), $NP$ (bright green), and $R$ (dark blue). Filled areas = daily rates of $R$new (light blue) and $GP$new (light green). Red dashed lines = framing the area of daily rates of $R$trad (below the zero line) and the area of $GP$trad (above the zero line). Solar noon is the time when the sun appears highest in the sky (maximum irradiance).

sets across various replicates per one species and per season to ensure enough data points for a reliable description of the general diurnal pattern. Data density for $R$ was comparatively low since the number of dark incubation periods per specimen was kept at a minimum during the day (3–4 per day), and because of occasional rate measurement ($NP$ and $R$) failures due to sporadic oxygen sensor spots malfunctions, particularly in summer. In summer, the model was therefore further restricted by applying a $k$-value of 6 to avoid overfitting of the data. Considering that $R$ and $NP$ are the same at sunrise and sunset when PAR is still below ~3 µmol photons m$^{-2}$ s$^{-1}$, the GAM for $NP$ was forced through the predicted value of $R$ at these points. This was achieved by adding predicted $R$ value at sunrise and at sunset to the raw $NP$ data sets and assigning a high weight (here 100) to them. A weighting function was then included in the $NP$ GAMs. Products of the GAM models were used to describe diurnal patterns of respiration, conduct comparisons between species, taxa, and seasons, define lag effects, and to calculate $GP$. The time lag between the peaks of hourly $R$ and $GP$ was determined and expressed in hours and minutes. Supplementary information is provided and includes files with the raw metabolic rates (.csv files), code of the GAMs including descriptions for .csv files (Script_for_GAMs.txt), predicted $R$ values at sunrise and set (Table S1 in S4 Data), and the GAM statistics (Table S2 in S4 Data).

The 95% confidence interval (CI) was calculated for each $R$ and $NP$ GAM. For derived $GP$ rates ($R + NP$), the CI was calculated by summing up their respective CIs ($CI_R + CI_{NP}$). The same was done when calculating daily rates: CIs were summed up in the same way as the hourly rates were summed up to calculate daily rates. The CI for daily $GP/R$ ratios (new and trad, respectively) was calculated as follows: $((CI_{GP}/GP) + (CI_R/R)) \times GP/R$.

## 3. Results

### 3.1 Environmental conditions

In fall, the mean temperature was 26.3°C during algae incubations and 25.5°C during coral incubations. In spring, the mean temperature was 21.3°C and in summer 29°C (Table 1). Daily PAR from aquarium measurements in fall was 7.25 ± 1.67 for algae incubations and 7.02 ± 1.00 mol photons m$^{-2}$ d$^{-1}$ for coral incubations. In spring, daily PAR was 12.55 ± 0.75 and in summer 10.09 ± 0.95 mol photons m$^{-2}$ d$^{-1}$ (Table 1). Integrated solar radiation levels above ground from the nearby Bermuda Weather Station were also reported, as it provides a better representation of the overall light regime in the respective seasons to which the study species were acclimatized. Seasonal variation of light was fairly similar between aquarium and weather station-based measurements, although somewhat higher inside the aquaria, likely due to differences in partial shading created by the experimental infrastructure (e.g., basin walls, roof scaffolding of the mesocosms; Table 1).

### 3.2 Diurnal pattern of respiration (*R*) in algae (fall) and corals (fall, spring, summer)

A clear diurnal pattern of *R* was evident, with a peak in the afternoon throughout all coral and algal species and seasons, with the exception of *Diploria* in summer (Fig 2). Note, that the high variability is not just due to intra-specific genetic variability in corals and algae, but also to differences in environmental conditions between days of measurements (weather/light), and the way algae rates were standardized (2D surface area). The general diurnal pattern was as follows: *R* was low at night, with the lowest rates often near sunrise. This was followed by a steady increase after sunrise, reaching a peak in the early or mid-afternoon. After the peak, *R* decreased fairly rapidly until sunset, then continued to decrease slowly until about midnight (Fig 2). In algae, the difference between minimum and maximum *R* varied 1.5- (*Caulerpa, Ceramium*) to 2.4-fold (*Laurencia*) based on the GAM-predicted diurnal pattern of *R* (Table S3 in S4 Data). In corals, this

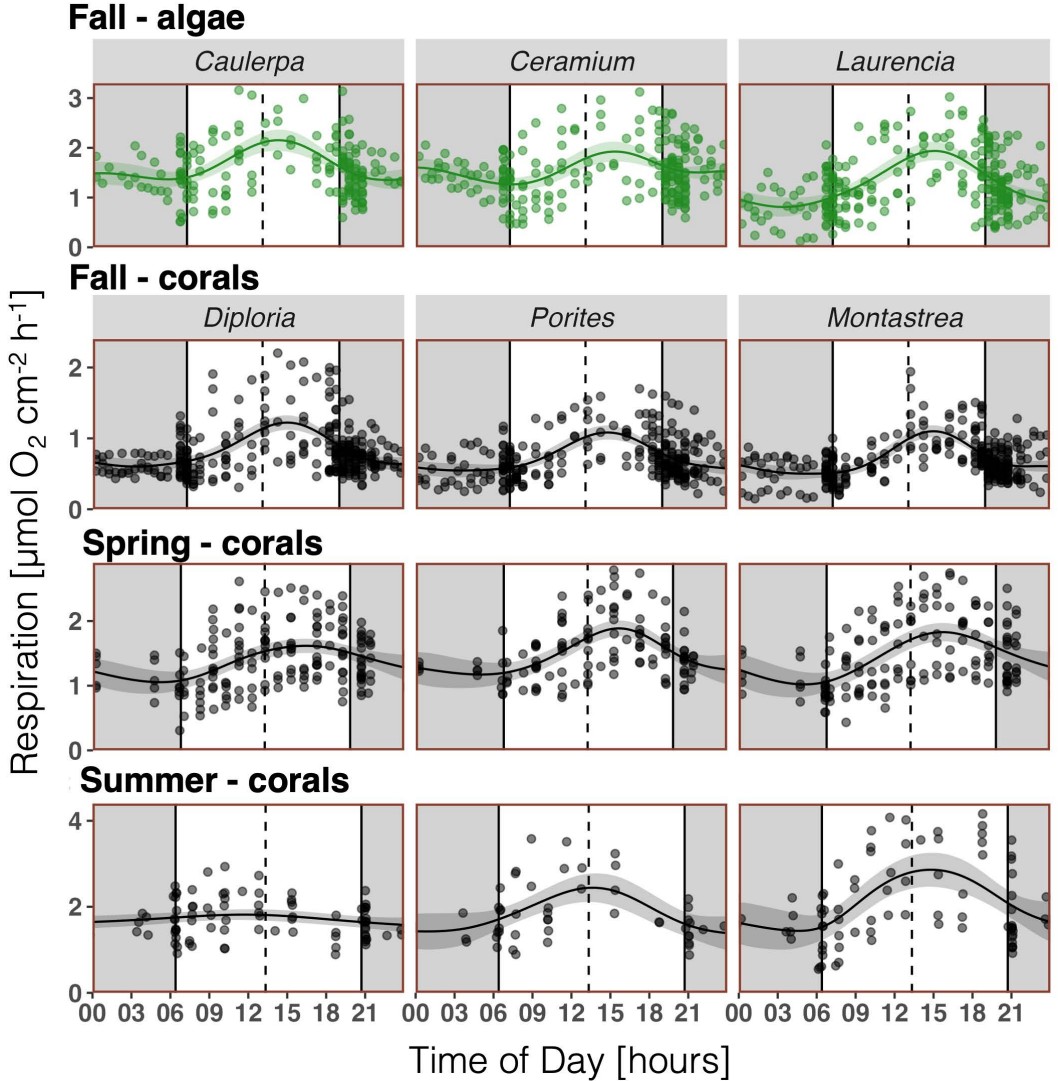

**Fig 2. Hourly respiration rates (*R*) of all experimental algae (green) and coral (black) specimens measured over several days per season.**
Algae (n = 9) and corals (n = 9) in fall 2020, corals (n = 7-9) in spring 2021, and corals (n = 7-9) in summer 2021. Night (dark background), solar noon (dashed line). *Curve fitting:* Generalized additive model (GAM) and 95% confidence interval.

difference varied between 1.2 and 2.2-fold across all species and seasons and averaged at 1.8-fold (Table S3 in S4 Data). All GAMs describing the diurnal patterns of $R$ were significantly different from a straight horizontal line (p<0.01), with the exception of *Diploria* in summer (p=0.185; Table S2 in S4 Data). The latter is due to a rather low data density for *Diploria* in summer, due to a number of sensor failures, which resulted in a weak curve fitting (Table S2 in S4 Data) and peak assessment.

Algae species had a higher $R$ than corals during the fall when both taxa were measured (Fig 3) with a maximum of $R$ of 2.08±0.21 µmol $O_2$ cm$^{-2}$ h$^{-1}$ for algal assemblages and 1.22±0.10 µmol $O_2$ cm$^{-2}$ h$^{-1}$ for corals (Table S3 in S4 Data).

If comparing $R$ of corals between seasons, it is evident that all corals had the highest $R$ rates in summer, with maximum hourly $R$ ranging between 1.82±0.12 (*Diploria*) and 2.86±0.39 µmol $O_2$ cm$^{-2}$ h$^{-1}$ (*Montastrea;* Fig 4, Table S3 in S4 Data). $R$ rates were lowest in fall, with maximum hourly $R$ ranging between 1.08±0.10 (*Porites*) and 1.22±0.10 µmol $O_2$ cm$^{-2}$ h$^{-1}$ (*Diploria;* Fig 4, Table S3 in S4 Data). The summer curves come with a rather high variability (large CI, Fig 4), which, again, is due to a comparatively low data density, as a number of incubations failed due to sensor failure.

### 3.3 Relating diurnal pattern of respiration ($R$) to photosynthesis ($NP$, $GP$)

The diurnal pattern of $NP$ and $GP$ closely follows light intensity as expected with a peak at solar noon (Fig 5). The raw data of $NP$ and respective GAM curve fittings are presented in the Supplementary Information (Fig S4-S7 in S4 Data). All GAMs were significantly different from a straight horizontal line (p<0.01; Table S2 in S4 Data). The peak of $R$ lagged behind the solar noon (Fig 2 and 3) and hence behind the peak of $NP$ and $GP$ (Fig 5).

This time lag ranged from minutes to hours, depending on species/ taxa and season. In algae (fall), the time lag varied between 1h:46min (*Ceramium*) and 2h:22min (*Laurencia*), which was longer than the time lag of corals in the same

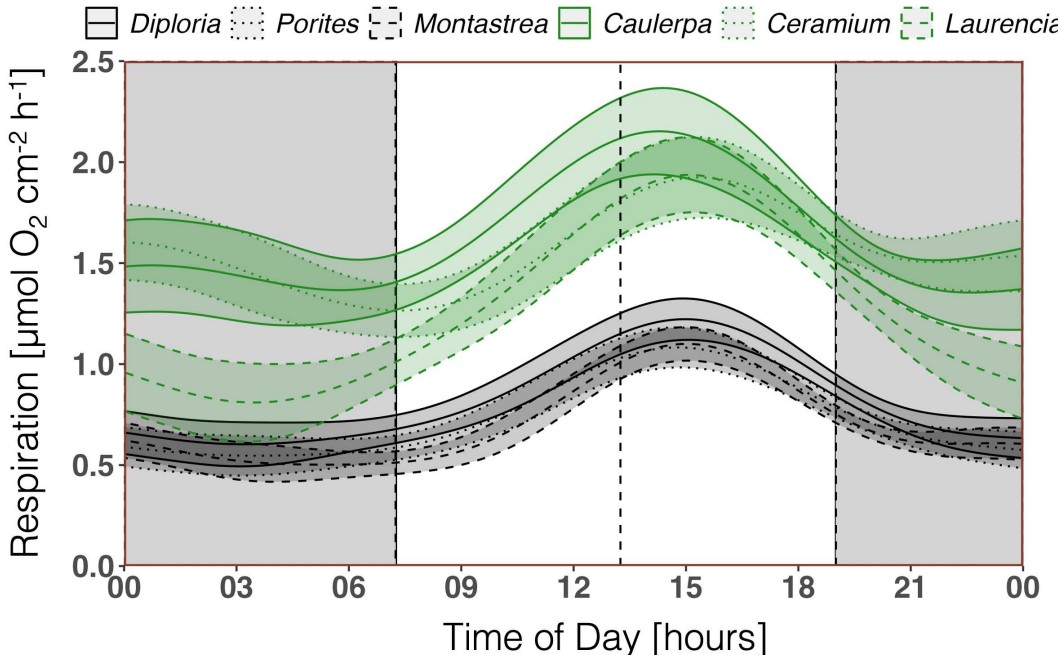

**Fig 3. Comparison of GAM-predicted diurnal patterns of respiration (*R*) between different algae (green) and coral species (black) in fall 2020.** Night (dark background), solar noon (dashed line). Mean±95% confidence interval.

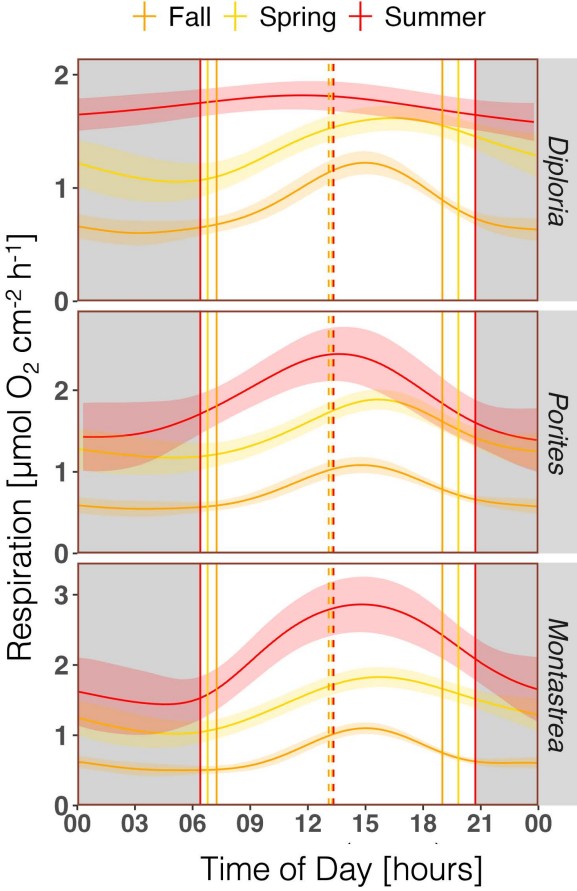

**Fig 4. Seasonal comparison of the diurnal pattern of GAM-derived respiration (*R*) of the three coral species.** Night (dark background), sunrise and sunset (solid vertical lines), solar noon (dashed lines). Mean ± 95% confidence interval.

season (1h:04 min [*Porites*] to 1h:43 min [*Diploria*]; Fig 6). Comparing coral time lags between seasons, the longest time lag was found in spring, ranging from 1h:35 min (*Montastrea*) to 3h:01 min (*Diploria*), and the shortest in summer, ranging from 0h:52 min (*Porites*) to 1h:22 min (*Montastrea*, Fig 6). As mentioned before, the diurnal pattern of *R* in *Diploria* in summer is non-significant (Table S1 in S4 Data), precluding reliable peak identification (Fig 2, 6).

### 3.4 Daily rates of respiration (*R*new) and gross photosynthesis (*GP*new) in algae (fall) and corals (fall, spring, summer)

Daily *R*new rates (integrated over 24 hours) of algae ranged between 30.7 ± 3.7 (*Laurencia*) and 38.9 ± 4.1 µmol cm⁻² d⁻¹ (*Caulerpa*) in fall, and daily *R*new rates of corals varied between 17.0 ± 1.7 (*Montastrea*, fall) and 49.2 ± 8.7 µmol cm⁻² d⁻¹ (*Montastrea*, summer; Fig 7, Table S3 in S4 Data). As for hourly maximum *R* rates, daily *R*new rates of corals were generally lowest in fall and highest in summer, with a 2.5-fold seasonal difference on average (Table S3 in S4 Data).

 *GP*new of algae varied between 38.4 ± 6.4 (*Ceramium*) and 50.1 ± 9.6 µmol cm⁻² d⁻¹ (*Caulerpa*) in fall, and in corals between 20.1 ± 3.3 (*Montastrea*, fall) and 67.9 ± 17 µmol cm⁻² d⁻¹ (*Porites*, summer), with the lowest rates in fall and highest rates in summer (Fig 7, Table S3 in S4 Data).

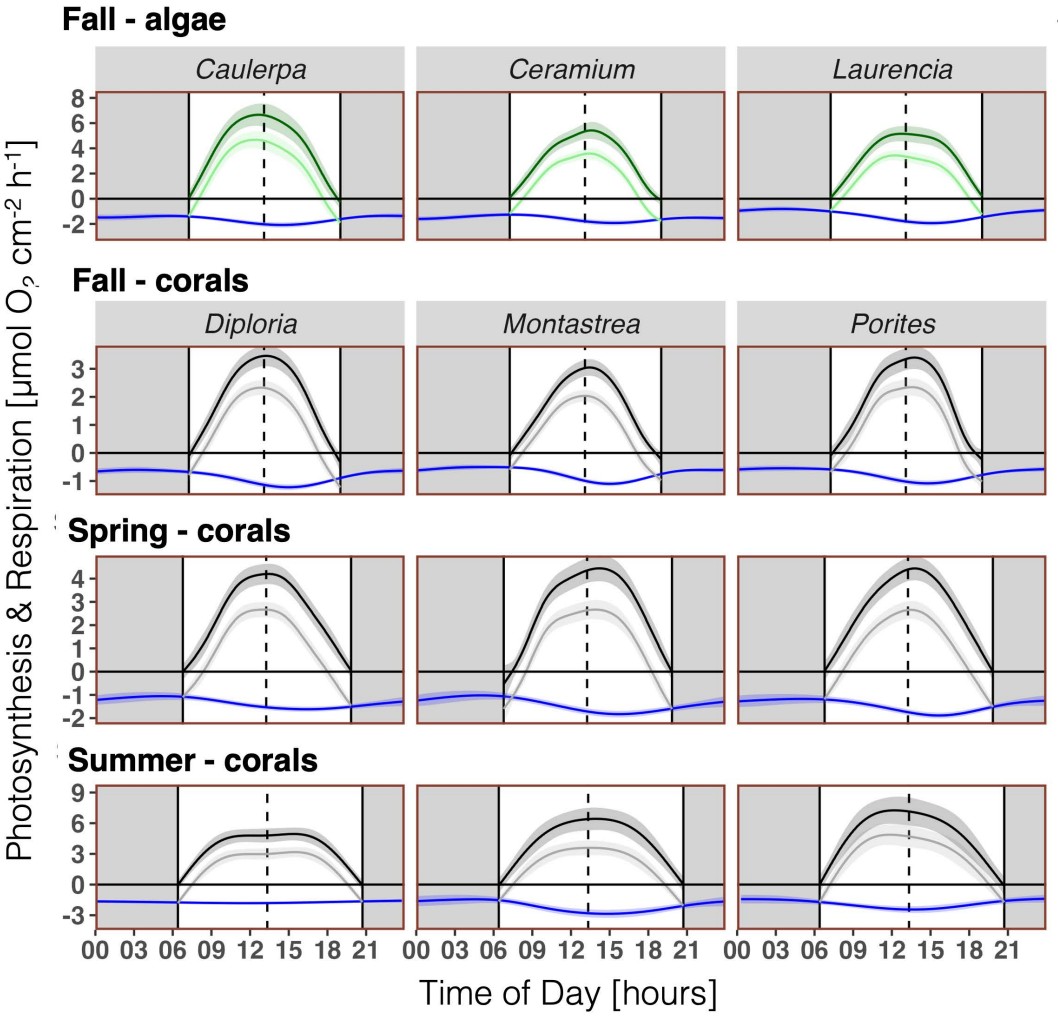

**Fig 5. GAM-predicted net photosynthesis (*NP*, light green or grey), gross photosynthesis (*GP*, dark green or grey), and respiration (*R*, blue) of algal species (green) in fall and coral species (grey) in fall, spring, and summer.** Night (dark background), solar noon (dashed line). Note: *R* is displayed as negative values (= negative oxygen evolution). Mean ± 95% confidence interval.

### 3.5 Comparing "new" and "traditional" daily respiration rates (*R*new versus *R*trad) and implications for daily gross photosynthesis rates (*GP*new versus *GP*new)

*R*new, which is calculated by incorporating the diurnal pattern of hourly R, was consistently higher than *R*trad, which assumes constant *R* throughout the day. The difference between the two was on average 14% across all algae and coral species (Fig 7). In fall, *R*new of algae was 5% (*Ceramium*) to 23% (*Laurencia*) higher than *R*trad, and *R*new of corals was 19% (*Montastrea*) to 20% (*Diploria*) higher. Comparing corals between seasons, the difference between *R*new and *R*trad was lowest in spring with a maximum of 13% (*Montastrea*, *Porites*) and highest in summer with up to 21% (*Montastrea*; Fig 7, Table S3 in S4 Data).

Differences in *R*new and *R*trad were consequently reflected in daily *GP*, where *GP*new was on average 13% higher than *GP*trad across all algae and coral species (Fig 7). In fall, *GP*new of algae was 5% (*Ceramium*) to 18% (*Laurencia*) higher than *GP*trad, and *GP*new of corals was 14% (*Porites*) to 17% (*Diploria*) higher. Comparing coral data between

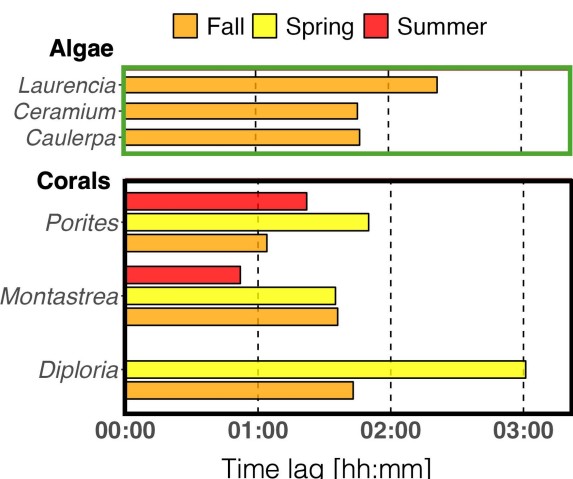

**Fig 6. Mean time lag between the GAM-derived peaks in respiration (*R*) and gross photosynthesis (*GP*) of algae in fall (top panel) and corals in fall, spring, and summer (bottom panel).** The time lag for *Diploria* in summer is missing as the GAM was non-significant, precluding peak identification.

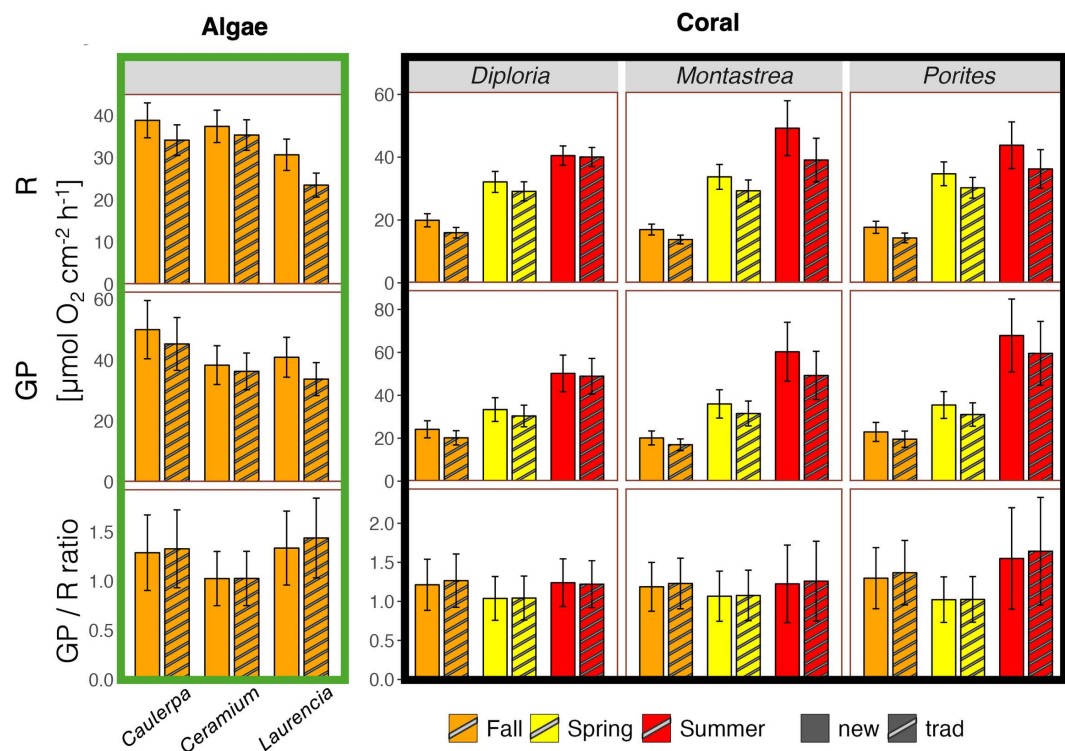

**Fig 7. Comparison of daily rates that consider diurnal patterns of hourly respiration ("new", filled bars) versus daily rates that are based on nighttime respiration only ("trad" = traditional, striped bars).** Daily respiration (*R*new and *R*trad) of all species and seasons in the top panel, daily gross photosynthesis (*GP*new and *GP*trad) in the center, and ratios of daily *GP* and *R* (*GP*new:*R*new and *GP*trad:*R*trad) in the bottom panel. Mean ± 95% confidence interval.

seasons, the relative difference between GPnew and GPtrad was lowest in spring with a maximum of 13% (*Montastrea, Porites*) and highest in summer with up to 18% (*Montastrea*; Fig 7, Table S3 in S4 Data).

The ratio of GPnew/Rnew was on average 1.21 across all species and seasons. They ranged between 1.03 ± 0.28 (*Ceramium*) and 1.34 ± 0.38 (*Laurencia*) in algae and between 1.18 ± 0.31 and 1.29 ± 0.39 in corals in fall (Fig 7). Comparing coral data between seasons, the ratios were lowest in spring with a maximum of 1.07 ± 0.32 (*Montastrea*) and highest in summer with a maximum of 1.55 ± 0.65 (*Porites,* Fig 7). Fall ratios were closer to summer values than to spring values (Fig 7). On average, the ratios derived from "new" rates (1.21) versus "traditional" rates (1.24) were very similar (Fig 7).

## 4. Discussion

The overarching goal of this study was to improve our understanding of coral and algae respiration and gross primary production under (near-) natural conditions. Most studies so far have used nighttime respiration rates (*R*trad) to calculate instantaneous or daily gross photosynthesis as well as daily respiration rates, which leads to an underestimation of energy/ carbon gain and utilization in corals and algae (e.g., [19–22,24–26]). The results of this study not only provide further evidence of this underestimation, but also uncover the variability and patterns of respiration diurnally and seasonally in important Western Atlantic coral and algal species.

The major outcomes are: (*i*) Pronounced diurnal patterns of respiration in all corals and algae investigated showing peak respiration rates in the afternoon – about 1–3 hours after the peak in photosynthesis – where respiration rates were on average 1.8 times higher in both, algae and corals compared to minimum respiration rates just before sunrise. (*ii*) Seasonal variation of respiration in all coral species, with the lowest respiration rates in fall, when PAR was also the lowest, and the highest respiration rates in summer, coinciding with highest temperature and PAR. The time-lag between the peak of photosynthesis and respiration was longest in spring and shortest in summer, following an inverse relationship with temperature. (*iii*) Daily respiration that considers daytime respiration (*R*new) was on average 14% higher in both algae (fall) and coral species (across all seasons) than daily *R* based on nighttime respiration only (*R*trad). This led to gross photosynthesis rates (*GP*new) that were on average 13% higher than traditional rates (*GP*trad), with the highest differences in corals found in summer (up to 18%). In the following, we will discuss these results in more detail, as well as the implications they have for calculating energy and carbon budgets.

### 4.1 Diurnal patterns in respiration

The increase of *R* in algae and corals in the morning following the increase in PAR and photosynthesis is expected, as studies previously found a positive relationship between the two [12,17]. However, *R* rates up to 2.4-fold higher at peak times in the afternoon compared to nighttime are higher than the findings of most previous studies that assessed relationships between light and respiration using the same (respirometry) approach. For example, Edmunds and Davies [12] found 1.6-fold higher respiration rates after 80 minutes at 140 µmol photons $m^{-2}$ $s^{-1}$ in the coral *Porites porites* collected at 10 m depth, which also plateaued at this rate with longer illumination time (120 minutes). Hoogenboom et al. [38] found 1.6-fold higher respiration rates in the coral *Tubinaria mesenterina* at the end of the afternoon compared to the early morning during a simulated diel light cycle in the laboratory with peak light intensities of 1200 µmol photons $m^{-2}$ $s^{-1}$. Lower differences were found during a diel cycle with lower light intensities (peak at 600 µmol photons $m^{-2}$ $s^{-1}$; [38]). Concerning algae, Tait and Schiel [39] found no significant differences between nighttime and daytime *R* of macroalgal assemblages in Southern New Zealand, although no data were shown to support this. It is unclear at what time of the day daytime *R* was measured, which was conducted by covering the chambers in-situ by a dark cloth. In phytoplankton, however, Falkowski et al. [16] demonstrated 1.3-fold and 1.6-fold elevated *R* rates in two different species after exposure to 700 µmol photons $m^{-2}$ $s^{-1}$ for only 5 minutes. And, Xue, Gauthier [17] found 1.8-fold (at 50 µmol photons $m^{-2}$ $s^{-1}$) to 2.8-fold (at 800 µmol photons $m^{-2}$ $s^{-1}$) higher oxygen consumption rates after 10 minutes of illumination in the unicellular green algae *Chlamydomonas reinhardtii*. While most of these studies found a positive relationship between respiration and light, none

of them investigated respiration rates over an entire diel cycle under natural light. This likely explains the discrepancy between our and previous results and highlights the importance of considering the natural dynamics of light, if the goal is to gain an understanding of in-situ coral and algae respiration rates and energy consumption.

A few things need to be considered when assuming that respiration rates measured in the dark immediately after a certain illumination period (commonly termed post-illumination dark respiration – LEDR) are the same as mitochondrial respiration during preceding light conditions, particularly in algae and plants. Conducting isotope measurements ($^{18}$O and $^{13}$C) during the light period and during the dark period after illumination, Xue et al. [17] found that while oxygen consumption is comparable in the light and the dark, $CO_2$ release is lower in the light than in the dark indicating lower $R$ during the light period. While it cannot be fully ruled out that some $CO_2$ might immediately be fixed again and therefore lead to an underestimation of $CO_2$ release under light conditions, it is also well known, that the TCA (or Krebs) cycle is suppressed during light conditions where most of the $CO_2$ is being produced [17,40–42]. At the same time, however, the activity of the mitochondrial electron transport chain is high under light (perhaps even stimulated by light – [16]) where $O_2$ is consumed and most of the ATP is being produced [17]. Therefore, we cannot say with certainty that energy (ATP) generated in mitochondria during LEDR is the same as during the light, but assuming that mitochondrial oxygen consumption rates are similar (following [17]), we can conclude that at least $GP$ calculations — the total amount of photosynthetic energy gain — is close to reality. Furthermore, in corals, the majority of the biomass is heterotrophic tissue of the coral host, which is likely unaffected by suppression of the TCA cycle under light. This is supported by a similar although less pronounced diurnal pattern of respiration in a cnidarian without zooxanthellae, the sea anemone *Nematostella vectensis,* with peak respiration rates in the afternoon being ~25% higher than during the night [43]. This was in contradiction to the authors' expectation, as *N. vectensis* is more physically active in the night, and led to the conclusion that this may be due to other energy-demanding processes occurring during the day, such as cell division and repair (e.g., oxygen radical scavenging; [43,44]).

### 4.2 Time lag and seasonality in respiration

The time lag between the peak in respiration and photosynthesis may be explained by two processes. First, more energy is likely required in the afternoon to counter potential damaging effects of light (scavenging of oxygen radicals and cell repair; [44]), which would be valid for both corals and algae. Second, in corals, the translocation of photosynthetic products to the coral host (e.g., glycerol and glucose) can take some time [45]. Interestingly, the time lag was shortest in summer, coinciding with highest PAR and temperature. While high light might require a more rapid response for protection and repair, higher temperature accelerates metabolic processes — including the translocation of photosynthetic products — and increases overall metabolic activity (i.e., highest respiration rates in summer; [10,46]). A time lag in corals was previously also observed between photosynthesis and calcification, with the highly energy-demanding calcification process peaking in the afternoon [47]. Contradictorily, however, the same study found a higher expression of genes involved in respiration in the night than in the daytime [47].

Respiration was lowest in fall and highest in summer, coinciding with seasonally changing PAR. This seasonality is partially in line with respiration measured in corals of the Gulf of Aqaba, Red Sea, where a similar seasonality in environmental parameters exists (latitude 29°N compared to Bermuda at 32°N). There, nighttime respiration (measured 1 hour after sunset) was highest in summer in all coral species investigated and lower in fall and spring [22]. While it is not possible to disentangle the effect of seasonally changing PAR and temperature in this study, as it investigated only three time points (seasons), it can be said with certainty that temperature also played a role in the overall lower respiration rates in fall and spring compared to summer [10].

### 4.3 Daily respiration, gross photosynthesis and energy budgets

Daily gross photosynthesis rates that incorporate the diurnal variation of respiration are on average 13% higher than traditionally calculated values that rely on night-time respiration only, with maximum variations of up to 18% in both, algae and

corals. This has considerable implications for coral energy and carbon budgets. Lesser [8] previously pointed out several shortcomings of coral energy budgets that rely on various assumptions that leave substantial space for errors. These include next to limited empirical data about biochemical composition, metabolic fuel preferences (lipids, carbohydrates, proteins), and metabolic quotients, a limited understanding of coral photosynthesis and respiration dynamics. Energy budgets typically rely on short-term measurements of instantaneous respiration and net photosynthesis to derive daily respiration and gross photosynthesis rates (as outlined in the introduction), which would mostly underestimate energy gain and carbon fluxes in corals. Conversely, overestimation is also possible if respiration rates measured immediately after a light period are extrapolated across the entire light period of a day.

The increased daytime respiration rates, primarily fueled by photosynthesis, indicate rapid and intense cycling of $CO_2$ and liable carbohydrates (e.g., glycerol, glucose) within the coral holobiont. Indeed, fixation and translocation of glucose have been found to occur within minutes in the sea anemone *Aiptasia* [48]. Similarly, light-induced accumulation of lipids (e.g., triacylglycerols, free fatty acids) in the zooxanthellae and the liposome of the coral host were found [49], with symbiont-host translocation rates of only 15 minutes [50]. While part of these accumulated compounds is utilized at night, it is very reasonable to assume that part of it facilitates increased respiration rates during the day [48]. Translocation of nitrogen-rich photosynthates, such as amino acids, takes longer, requiring 3 hours or more [50].

The acquired energy may be directly utilized for cell maintenance and to combat oxidative stress caused by high light and UV [51,52] as well as to facilitate energy-consuming calcification [53–55] and mucus production [56], which are both known to be higher in the daytime. Note that it has also been suggested that not photosynthesis, but light directly enhances calcification [57]. In algae, energy expenses at daytime are similar to those in corals, such as antioxidant activity [58,59], higher growth during the daytime [60], and a higher release of dissolved organic carbon [61,62], which can include bioactive compounds [63].

When calculating energy budgets, the utilization of energy is typically separated into two main categories: maintenance and growth. The so-called scope-for-growth (SfG) is defined as the difference between energy acquisition and that lost via respiration and excretion [64], a concept that has been applied several times in corals (e.g., [4,65–67]). The photosynthetically-derived energy that is available for growth is therefore the difference between daily gross photosynthesis and respiration, minus excretion, such as mucus release. In this study, we did not assess mucus release; therefore, we can only differentiate between energy used (i) for maintenance and (ii) for growth + excretion. Coming back to the comparison of the "new" and the "traditional" approach of calculating daily respiration and gross photosynthesis, we found that the gross photosynthesis/respiration (*GP/R*) ratio was hardly affected by the method of calculation, which is to be expected as gross photosynthesis increases with an increase of respiration. This means that both approaches capture the amount of energy used for growth + excretion similarly well. However, the calculated amount of energy used for maintenance (= energy turnover) is higher in the "new" approach, indicated by a higher energy acquisition (*GP*new) and a higher utilization (*R*new). Those turnover rates reflect the scale of metabolic activity, including, for example, energy requirements of transmembrane transporters, movement, and cellular maintenance processes such as protein turnover and repair mechanisms. Quantifying these rates is important because their demand can vary and adjust in response to changing environmental conditions. Species or genotypes with high energy turnover—and/or a greater capacity to regulate their energy turnover in response to changing demands—may be better equipped to cope with environmental stressors such as ocean warming and acidification. For instance, some corals can increase the activity of energy-demanding $H^+$-pumps at the site of calcification to maintain elevated pH levels in the calcifying fluid, even when seawater pH decreases [68,69]. So far, energy turnover rates remain poorly described, which limits our ability to understand and predict the capacity of corals to adjust and respond to environmental change.

In corals, quantifying energy budgets is further complicated by the fact that they also gain energy by heterotrophy. While the uptake of energy (/carbon) by photosynthesis can be quantified comparatively easily, the quantification of total heterotrophic uptake is challenging, in particular, if the various heterotrophic sources such as dissolved organic matter,

detritus, pico- and micro-plankton, and mesozooplankton are taken into account [70]. Indeed, while various studies investigated the energy and nutrient uptake of one or few heterotophic sources (e.g., [71–73]), we are currently unaware of a study that investigated all sources simultaneously. Heterotrophic input can be very dynamic depending on the availability of food sources and water flow, and on the demand, which can increase during stressful conditions [70,72,74]. As quantifying heterotrophic input is challenging, it should theoretically be possible to calculate heterotrophic energy gain if carbon turnover and investment into growth (tissue and skeleton) and mucus (dissolved and particulate organic carbon) can be quantified reliably. Here, accurate quantification of respiration throughout the day and accurate calculation of gross photosynthesis are crucial.

In conclusion, this study provides significant advances in understanding coral and algae respiration and gross primary production under near-natural conditions by revealing critical diurnal variations that challenge traditional methods of energy budget calculations. Respiration rates were highest in early- to mid-afternoon, linked to photosynthetic activity and increased energy demands for cell operations and maintenance. Traditional methods (based on extrapolation of night-time respiration rates) underestimated photosynthetic energy gain and energy utilization (respiration) by up to 18%. As the need for cell operations and maintenance varies with changing environmental conditions, stressful and non-stressful, and likely also during different life stages of the coral (and algae), it is important to quantify respiration and gross photosynthesis accurately to draw conclusions on phenotypic plasticity. This refined approach holds implications for predicting organism resilience to changing environmental conditions more accurately, as resilience is directly linked to energy availability, energy allocation, and turnover.

## Supporting information

**S1 Data.  my_data_resp.csv – contains raw hourly respiration rates. fall_NP_coral.csv** – contains raw hourly net photosynthesis rates of corals in fall. **fall_NP_algae.csv** – contains raw hourly net photosynthesis rates of algae in fall. **spring_NP_coral.cvs** – contains raw hourly net photosynthesis rates of corals in spring. **summer_NP_coral_v2.csv** – contains raw hourly net photosynthesis rates of corals in summer.
(ZIP)

**S2 Data.  Script_for_GAMs.txt – contains the scripts for the generalized additive models (GAMs) used to describe diurnal pattern of respiration (*R*) and net photosynthesis (*NP*).**
(TXT)

**S3 Data.  Inclusivity in global research.**
(DOCX)

**S4 Data.  Supporting Information_methods and results – contains both the supporting tables and figures.**
(PDF)

## Acknowledgments

We are grateful for the assistance of the BIOS interns Chloe Root and Benjamin Shirey, who conducted measurements in summer 2021 supported by the Galbraith/Wardman Fellowship. We also want to thank the personnel of the Bermuda Weather Station for kindly providing the solar insulation data.

## Author contributions

**Conceptualization:** Yvonne Sawall.

**Data curation:** Roderick Bakker, Natalia E. Padillo-Anthemides, Nicole Adamson.

**Formal analysis:** Yvonne Sawall, Roderick Bakker.

**Funding acquisition:** Yvonne Sawall, Roderick Bakker.

**Investigation:** Yvonne Sawall, Roderick Bakker.

**Project administration:** Yvonne Sawall.

**Supervision:** Yvonne Sawall.

**Visualization:** Roderick Bakker.

**Writing – original draft:** Yvonne Sawall.

**Writing – review & editing:** Roderick Bakker, Natalia E. Padillo-Anthemides, Nicole Adamson.

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
