## [Decision Letter · Decision Letter 0]

PONE-D-24-55887Diurnal pattern of respiration in corals and algae and its implications for gross primary production quantification

PLOS ONE

Dear Dr. Sawall,

Thank you for submitting your manuscript to PLOS ONE. After careful consideration, we feel that it has merit but does not fully meet PLOS ONE’s publication criteria as it currently stands. Therefore, we invite you to submit a revised version of the manuscript that addresses the points raised during the review process.

We look forward to receiving your revised manuscript.

Kind regards,

Claudia Isabella Pogoreutz

Academic Editor

PLOS ONE

Journal Requirements:

 “YS supported by BIOS Cawthorn Innovation Fund project (2019-2021).

RB supported by the Erasmus program for student exchange.

NPA and NA supported by the National Science Foundation REU Program at BIOS, Award# OCE-1757475”

5. Please note that your Data Availability Statement is currently missing the repository name and/or the DOI/accession number of each dataset OR a direct link to access each database. If your manuscript is accepted for publication, you will be asked to provide these details on a very short timeline. We therefore suggest that you provide this information now, though we will not hold up the peer review process if you are unable.

Additional Editor Comments:

The reviewers' main comments and suggestions include:

-    Additional analyses appear for specific subsets of data (comparison of measured oxygen fluxes between new and traditional methods);

-    Additional supplementary data to support certain statements in the manuscript;

-    A schematic representation of the methodology (for details, please refer to detailed comments below);

-    More detailed discussion and research recommendations based on Rnew - specifically: what are the implications of potentially higher turnover rates in algae and corals (e.g., for energy budgets);

-    Suggestions to improve color schemes of specific figures

-    Code should be made available (e.g., publication on github or similar repository).

Reviewers' comments:

Reviewer's Responses to Questions

**Comments to the Author**

1. Is the manuscript technically sound, and do the data support the conclusions?

Reviewer #1: Yes

Reviewer #2: Yes

2. Has the statistical analysis been performed appropriately and rigorously? 

Reviewer #1: I Don't Know

Reviewer #2: Yes

3. Have the authors made all data underlying the findings in their manuscript fully available?

Reviewer #1: Yes

Reviewer #2: No

4. Is the manuscript presented in an intelligible fashion and written in standard English?

Reviewer #1: Yes

Reviewer #2: Yes

5. Review Comments to the Author

Reviewer #1: This nicely written and concise manuscript mainly aims at unraveling diurnal and seasonal patterns in coral and algae respiration rates. For this, three coral species and three algal species were collected and respiration (and photosynthesis) rates measured at three different timepoints throughout the year. A new way was established to calculate respiration rates, by calculating daily rates instead of assuming night respiration to be constant, and subsequently gross photosynthesis and GP:R ratios. The manuscript finds that traditional rates may underestimate daily R and GP rates, highlighting a more temporal resolution of diurnal changes in oxygen fluxes.

The manuscript is well written and focusing on the main aspects of the work. While the introduction is very concise and to the point, I am missing some references in the text (as can be seen in my detailed comments below). The methods are described well and allow for reproducibility of the setup, though I am missing a bit more detail regarding the incubation time for net photosynthesis rates and the amount of run controls. The results are really nice and supported by well executed supporting figures and tables, including the supplementary material. I am missing statistical tests though to see if the reported differences in oxygen fluxes between the new and traditional method are significant or not, as based on Figure 6 it visually looks like the results may not be significant. The discussion is also written well, missing maybe some references, but I am missing more detail and recommendation in the discussion about Rnew specifically. The study is definitely interesting and has a good basis, but I think by addressing below comments it can be improved. The comments are just ordered by line numbers.

27-32; 34-38; 40-42; 46-52; 110-112: no references given for those statements

77-79: did you mean to say “provides R rates closest to the actual GP”? otherwise I am unsure what you want to say about R rates in this kind of experimental setup

113-116: Does this mean the rocks were also overgrown with other algae and/or organisms beside the dominant algae of interest? Please specify. And if there were also other organisms, besides the algae of interest, on those rocks please elaborate how you accounted for that and how this might influence the measured fluxes.

Table 1: as the temperature range is quite high, maybe it would be nice to also indicate the mean temperature for the incubation periods (mean temperatures are described in the text line 233-234)

149-151: was there a lack of hobo loggers to put one in each incubation chamber? Or was there another reason (e.g. sufficient replication) so only 7 incubation chambers were continuously monitored?

161-163: Can the authors please supply the data and analysis to support this claim in the supplementary material? Also, was there a comparison done between potential differences of oxygen flux measurements based on continuous measurements (fall) vs 30s measurements at beginning and end (summer)? Lastly, why was there a change of methods in oxygen flux measurements within the same study (would be interesting to know if comparing methods was a planned part of the study, or if it had different reasons)?

167-169: It is unclear to me how long the specific light and dark incubations were run for. The dark incubations apparently always lasted 30 min (based on line 175), but then the light incubation was how long? And were the light incubations done before or after the respiration measurements?

181-182: how many controls were run in total? Or if I understood the setup correctly each day 3 corals and 3 algae were tested, leaving space in 3 incubation chambers for controls on each day. Did the authors run 3 controls each day?

185-190: It would be interesting to know on how many pictures the 3D models relied on (e.g around 100 pictures per colony/rock); also was there any conversion used to account for the structure of the algae growing on the rocks? For example the study of Cardini et al. 2016 used a conversion factor of 3.9 for turf algae surface area measurements (https://doi.org/10.1007/S10021-016-9966-1) and maybe this would be of interest/relevant for the algae species studied here.

194: missing closing brackets after the reference

191-202: I think the calculations would benefit from a visual representation of how they were done, for the readers understanding of the process. Like a schematic representation of the curves, hourly rates, and which rates where summed for R/GPnew vs R/GPtrad calculations for a single incubation chamber over the course of 24h incubations

240-243: as the units between PAR and solar radiation at the weather station are different it is hard for the reader to compare if the values are actually similar or not. Maybe it would be an idea to convert them into the same unit to have a clearer comparison.

302-304: I am not sure if the slope of the declining curve in the afternoon is the main/only characteristic to determine how pronounced the diurnal pattern is. What about the range/amplitude of the curve? If you compare the range of R of Porites between spring (0.72) and fall (0.53), or for Montastrea (spring: 0.81; fall: 0.6) the amplitude is higher in spring, so I would argue that could show a more pronounced diurnal pattern in spring (contrary to what the authors are saying). Though both arguably have a way lower amplitude in spring and fall compared to summer, so maybe this alternative/additional comparison is not as fruitful as I was initially assuming.

312-313 and throughout the results: When talking about daily R rates – as Rnew and Rtrad were introduced in the beginning, please always write which R rates are talked about in the results specifically, so the reader knows.

318-339: Did the authors test if the reported differences are statistically significant? This analysis would help strengthening their point, rather than just reporting the percent difference. This goes for the comparison specifically of Rnew to Rtrad (and generally all derived values - R, GP, and GP/R – and their new and trad comparison), but could also be done for comparing the difference in GP or GP/R between the different months if this would be of interest. Statistical testing could really boost the paper, and would needed to be included in the Material and Methods and Results.

351-354; 420-422; 424-427; 486-489: no references given for those statements

375-377: While the authors are nicely highlighting the previous studies and how their results differ from the increased R rates measured in this study (lines 377-392), I am lacking an explanation or implication of the findings of this study. As the findings are contrary to previous ones how can this difference be explained? Do the new results have specific implications?

408: missing closing brackets after the reference

480-489: I am missing more detail and recommendation in the discussion about Rnew and what these potentially higher turnover rates in algae and coral mean. How does this affect current knowledge about energy budgets, etc.?

514-517: missing more detail about this conclusion in the discussion, as it is never really discussed in light of previous literature.

Reviewer #2: Answers to the questions above.

1. The manuscript presents results of original research and describes a technically sound piece of scientific research. The experiments were performed with a sufficient amount of replicates and controls.

2. To the best of my knowledge and the presented data, the statistical analysis has been performed appropriately.

3. Not all data and data analysis steps are made available. In the Supplementary only the summary output of the GAM model and the summary of the metabolic rate can be found. It is not clear whether or not these tables contain all the data that was collected or just an overview. More information in the table description would be useful to clarify this. Furthermore, a Github site containing all of the data as well as the data sheets and codes used for the analysis should be provided.

4. Yes, the manuscript is written in a standard English.

5. Additional comments

1. The beginning of the introduction is not well structured. The first paragraph (line 27-32) of the introduction reads very miss placed. I would move this paragraph down one paragraph and start with a more general introduction first and then go into detail about Respiration and photosynthesis and the current state of knowledge regarding the R and PS dynamics.

2. In the material and method part, the sections are not well organized. E.g. Line116-127 rather experimental design than study species. Line 148-155 rather experimental design than metabolic rate measurement, Line 163-165 rather experimental design than metabolic rate measurement. Line 178-179 rather metabolic rate calculation than measurement and also repetition to line 191.

3. Table1 (line 129-132) and figures: The units should be written in []

4. The quality of all figures have to be improved. The colors used in the figures are not intuitive.. Furthermore, it is confusing that NP and spring have the same color. The figures could be improved by choosing colors such as: Dark green GP, light green NP, blue R and for the seasons: fall: orange, spring: yellow, summer: red. The figure descriptions have to be extended to a degree that the figure and the description can stand alone to understand the plot. Therefore, I would recommend to add more information and also the main result that is shown in the plot as the title of the figure.

5. The result section could also be structured better, by dividing the metabolic rates section into diurnal and season or by respiration and photosystheis results. As title for each section the main finding should be used to clearly stating the result.

6. Was the symbiont density also investigated? In line 267 you mention that: "Algal assemblages had a higher R as corals during the fall .." Is this only due to higher metabolic efficiency of the algae compared to the coral or could it also be due to differences in algae density compared to the coral? the difference between algae and coral would also be interesting to discuss in the discussion section.

7. The explanation of higher R in line 377 to 392 is very extensively described by stating the exact results of other studies and there measured values. This section could be shortened by focusing on one example specifically and only briefly mentioning the others or by focusing on the main differences and discussing it in comparison to this study.

6. PLOS authors have the option to publish the peer review history of their article (what does this mean? ). If published, this will include your full peer review and any attached files.

**Do you want your identity to be public for this peer review?** For information about this choice, including consent withdrawal, please see our Privacy Policy .

Reviewer #1: **Yes: ** Selma D. Mezger

Reviewer #2: No

---

## [Author Response · Author response to Decision Letter 1]

14 Apr 2025

All comments of the reviewers and the editor have been addressed in the rebuttal letter.

---

## [Decision Letter · Decision Letter 1]

PONE-D-24-55887R1Diurnal pattern of respiration in corals and algae and its implications for gross primary production quantificationPLOS ONE

Dear Dr. Sawall,

Thank you for submitting your manuscript to PLOS ONE. After careful consideration, we feel that it has merit but does not fully meet PLOS ONE’s publication criteria as it currently stands. Therefore, we invite you to submit a revised version of the manuscript that addresses the points raised during the review process.

 One of the reviewers has highlighted few remaining formatting and design issues with the figures, which I believe can be quickly revised. I am looking forward to receiving a final version of your manuscript.

We look forward to receiving your revised manuscript.

Kind regards,

Claudia Isabella Pogoreutz

Academic Editor

PLOS ONE

Journal Requirements:

Reviewers' comments:

Reviewer's Responses to Questions

**Comments to the Author**

1. If the authors have adequately addressed your comments raised in a previous round of review and you feel that this manuscript is now acceptable for publication, you may indicate that here to bypass the “Comments to the Author” section, enter your conflict of interest statement in the “Confidential to Editor” section, and submit your "Accept" recommendation.

Reviewer #1: All comments have been addressed

Reviewer #2: All comments have been addressed

2. Is the manuscript technically sound, and do the data support the conclusions?

Reviewer #1: Yes

Reviewer #2: Yes

3. Has the statistical analysis been performed appropriately and rigorously? 

Reviewer #1: I Don't Know

Reviewer #2: Yes

4. Have the authors made all data underlying the findings in their manuscript fully available?

Reviewer #1: Yes

Reviewer #2: Yes

5. Is the manuscript presented in an intelligible fashion and written in standard English?

Reviewer #1: Yes

Reviewer #2: Yes

6. Review Comments to the Author

Reviewer #1: The authors have answered all my questions and comments fully in a nice, understanding, and concise way. I fully support this manuscript to be published.

I only found 3 minor formatting "issues" that should be corrected in editing:

Line 202: "were" different font

Fig. 2 description: different font sizes

Fig. 3, 4, and 5 description: "confidence interval" in different font size

Thanks for the opportunity to peer review this work.

Reviewer #2: 1) The authors have adequately addressed the comments raised by reviewer 1 and 2 and incorporated the highlighted the raised comments well into the revised manuscript.

2) The manuscript is now more technically sound and the data is now better presented and explained.

3) The statistical analysis has been performed appropriately and the data description and presentation is now improved by the additional explanations and changes in the manuscript.

4) The underlying data is now better accessible and fully available.

5) yes, but the manuscript should be checked again for grammatical error since some minor errors can be found e.g. line 301.

6) additional comments

- In the description of fig.1 the font and size of the text changes.

- In figure 5 the GP and NP of the corals plots is not correctly colored. Adjust color legend or adjust figure.

- The quality of the figures is still quite bad. Please improve for final manuscript.

- In the new figure 1 the unit of the y-axis is not provided, please make sure that units are presented in squared brackets [].

7. PLOS authors have the option to publish the peer review history of their article (what does this mean? ). If published, this will include your full peer review and any attached files.

**Do you want your identity to be public for this peer review?** For information about this choice, including consent withdrawal, please see our Privacy Policy .

Reviewer #1: **Yes: ** Selma D. Mezger

Reviewer #2: No

---

## [Author Response · Author response to Decision Letter 2]

23 May 2025

Reviewer #1: The authors have answered all my questions and comments fully in a nice, understanding, and concise way. I fully support this manuscript to be published.

I only found 3 minor formatting "issues" that should be corrected in editing:

Line 202: "were" different font

RE: Fixed. Can’t believe that you even found this :)

Fig. 2 description: different font sizes

RE: Fixed.

Fig. 3, 4, and 5 description: "confidence interval" in different font size

RE: Fixed.

Reviewer #2: 1) The authors have adequately addressed the comments raised by reviewer 1 and 2 and incorporated the highlighted the raised comments well into the revised manuscript.

2) The manuscript is now more technically sound and the data is now better presented and explained.

3) The statistical analysis has been performed appropriately and the data description and presentation is now improved by the additional explanations and changes in the manuscript.

4) The underlying data is now better accessible and fully available.

5) yes, but the manuscript should be checked again for grammatical error since some minor errors can be found e.g. line 301.

RE: We did a thorough read through the ms again and fixed a few minor errors.

6) additional comments

- In the description of fig.1 the font and size of the text changes.

RE: Fixed.

- In figure 5 the GP and NP of the corals plots is not correctly colored. Adjust color legend or adjust figure.

RE: We removed the color legend and described the plots and colors in the figure caption.

- The quality of the figures is still quite bad. Please improve for final manuscript.

RE: We did some further improvements to the quality of the figures.

- In the new figure 1 the unit of the y-axis is not provided, please make sure that units are presented in squared brackets [].

RE: We added the units in square brackets. We also modified the other figures in terms of units in squared brackets.

---

## [Editor Report · Decision Letter 2]

Diurnal pattern of respiration in corals and algae and its implications for gross primary production quantification

PONE-D-24-55887R2

Dear Dr. Sawall,

We’re pleased to inform you that your manuscript has been judged scientifically suitable for publication and will be formally accepted for publication once it meets all outstanding technical requirements.

Kind regards,

Claudia Isabella Pogoreutz

Academic Editor

PLOS ONE
---

## [Editor Report · Acceptance letter]

PONE-D-24-55887R2

PLOS ONE

Dear Dr. Sawall,

I'm pleased to inform you that your manuscript has been deemed suitable for publication in PLOS ONE. Congratulations! Your manuscript is now being handed over to our production team.

Kind regards,

on behalf of

Prof. Claudia Isabella Pogoreutz

Academic Editor

PLOS ONE